



# NorthSEAL: A new Dataset of Sea Level Changes in the North Sea from Satellite Altimetry

Denise Dettmering[1], Felix L. Müller[1], Julius Oelsmann[1], Marcello Passaro[1], Christian Schwatke[1], Marco Restano[2], Jérôme Benveniste[3], and Florian Seitz[1]

[1]Deutsches Geodätisches Forschungsinstitut der Technischen Universität München (DGFI-TUM), Arcisstrasse 21, 80333 Munich, Germany
[2]SERCO, c/o ESRIN, Largo Galileo Galilei, Frascati, Italy
[3]European Space Agency (ESA-ESRIN), Largo Galileo Galilei, Frascati, Italy

**Correspondence:** Denise Dettmering (denise.dettmering@tum.de)

**Abstract.** Information on sea level and its temporal and spatial variability is of great importance for various scientific, societal and economic issues. This article reports about a new sea level dataset for the North Sea (named NorthSEAL) of monthly sea level anomalies (SLA), absolute sea level trends and sea level mean annual amplitudes over the period 1995-2019. Uncertainties and quality flags are provided together with the data. The dataset has been created from multi-mission cross-calibrated altimetry data, preprocessed with coastal dedicated approaches and gridded with innovative methods to a 6-8 km wide triangular mesh. The comparison of SLA and tide gauge time series shows a good consistency with average correlations of 0.85 and maximum correlations of 0.93. The improvement with respect to existing global gridded altimetry solutions amounts to 8-10%, and it is most pronounced in complicated coastal environments such as river mouths or regions sheltered by islands. The differences in trends at tide gauge locations depend on the vertical land motion model used to correct relative sea level trends. The best consistency with a median difference of 0.04 ± 1.15 mm/year is reached by applying a recent glacial isostatic adjustment (GIA) model. With the presented sea level dataset, for the first time, a regionally optimized product for the entire North Sea is made available. It will enable further investigations of ocean processes, sea level projections and studies on coastal adaptation measures. The NorthSEAL data is available at https://doi.org/10.17882/79673 (Müller et al., 2021).

## 1 Introduction

Sea level is one of the essential climate variables (ECV) as defined by the Global Climate Observing System (GCOS), and sea level rise is one of the most discussed topics in the context of global change. Risk assessment of potential threats along the coasts by rising sea levels in connection with extreme events requires a solid data basis of sea level changes over the past and predictions of its future evolution. A rise of the mean sea level (MSL) is accompanied by a higher probability of severe storm surges and floods (Wahl, 2017). Comprehensive and long time series of precise sea level observations are thus decisive





for the development of appropriate adaptation measures. Furthermore, high quality observation data of sea level provides a valuable contribution to the general understanding of interactions and processes in the climate system. The coastal regions of the North Sea are in parts densely populated and of great economic significance. Especially for low-lying areas along large coastal stretches of the German Bight, coastal protection measures, such as dike building, are of paramount importance and

associated with great efforts (Sterr, 2008).

The North Sea area is well equipped with measurement systems monitoring sea level and its changes. Along the coastlines many tide gauge (TG) stations provide valuable data, in some cases already for more than 100 years. In addition, satellite altimetry can be used to monitor sea level offshore. Even if these time series are only about 25 years long, they are homogeneously distributed over the entire North Sea and provide the water stage in an absolute sense - in contrast to TG readings,

which are referenced to fixed points on land, and are prone to vertical land motion (VLM). However, the temporal resolution of satellite altimetry is quite sparse, and the creation of long-term and high-resolution sea level information requires the combination of different satellite missions. Moreover, especially in the vicinity of coasts where land and calm water may influence the radar echoes, the observation data needs to be carefully pre-processed.

Today, a few global altimetry-based sea level datasets are available. One of them has been developed in the frame of the
ESA Sea Level Climate Change Initiative (SL_cci) (Legeais et al., 2018), and another one is an operational product, computed by the Data Unification and Altimeter Combination System (DUACS) and provided by the Copernicus Marine Environment Monitoring Service (CMEMS) and the Copernicus Climate Change Service (C3S) (Taburet et al., 2019). Global products include the North Sea, but they are neither optimized for regional nor for coastal applications. Regional products from SL_cci covering the North Sea have recently shown enhanced coastal capabilities (Birol et al., 2021; Benveniste et al., 2020), but they
are limited to along-track analysis of selected missions.

Most studies investigating MSL changes in the North Sea are based on country-wide TG analyses (e.g. Woodworth et al. (2009), Albrecht et al. (2011), Richter et al. (2012)). In contrast, Shennan and Woodworth (1992) and later Wahl et al. (2013) use long-term sea level measurements from a set of 30 TG covering the entire North Sea coastline. Not surprisingly, the detected sea level trends as well as the inter-annual variability are not uniform along the coastline. For the period 1900-2011
and for the whole North Sea, Wahl et al. (2013) found an absolute MSL trend of $1.53 \pm 0.16$ mm per year, which increases to $4.00 \pm 1.53$ mm per year when only taking the period 1993-2009 into account. Later, Dangendorf et al. (2014) extended this study to investigate the intra-annual to decadal-scale sea level variability in order to better understand the underlying processes. More recently, Frederikse and Gerkema (2018) demonstrated that the low-frequency variability in the seasonal deviations from annual mean sea level is mostly driven by wind and pressure. Information on open ocean areas was not derived in any of theses
studies, since this cannot be obtained from TG measurements. This is a critical limitation, since from Benveniste et al. (2020) and Gouzenes et al. (2020) it is known, that coastal sea level trends cannot always be transferred to offshore regions. Moreover, sea level changes differ significantly from region to region (Stammer et al., 2013).

Observation data from satellite altimetry to monitor open ocean sea level variations is available since 1992. An early study using these data in the North Sea was published by Høyer and Andersen (2003). They assessed data from the TOPEX/Poseidon
mission and compared it with TG data with the aim of assimilating both data types into storm surge models. Already at that



time, they found the Root Mean Square error (RMS) of merged altimetry and TG data to be significantly lower than for the models. More recently, Sterlini et al. (2017) analyzed satellite altimetry data to assess the causes for spatial variability of sea level trends in the North Sea. Within their study, they were able to address the spatial nature of the physical mechanisms that are responsible for sea level change due to the availability of observation data over the open ocean. The altimetry data used was extracted from a global dataset provided by AVISO - most probably DUACS DT2014 (Pujol et al., 2016). Due to the lack of coastal dedicated altimeter data pre-processing they focused on offshore regions only.

In recent years, the quality and quantity of altimetry data in the coastal zone have improved considerably (Cipollini et al., 2017): advanced radar waveform processing techniques have been developed (e.g. ALES retracker (Passaro et al., 2014)), coastal-dedicated geophysical corrections are available (e.g. GPD+ troposphere correction (Fernandes and Lázaro, 2016)), and innovative altimeter instruments are providing data (e.g. SAR altimetry from Sentinel-3).

In this study, a new gridded altimetry-based regional sea level dataset for the North Sea is presented, named NorthSEAL. It is based on long-term multi-mission cross-calibrated data, consistently pre-processed with coastal dedicated algorithms and gridded to a 6-8 km wide triangular mesh with innovative methods. The dataset enables advanced region-wide sea level studies and investigations of ocean processes causing sea level variations on different spatial and temporal scales.

The paper is structured as follows: Sect. 2 introduces the study area and the used input and validation data. Sect. 3 describes the methods applied for data pre-processing, gridding, and estimating derived parameters, i.e. trends and annual amplitudes. Sect. 4 provides detailed information on the resulting dataset, such as time period, resolution and data format, and Sect. 5 discusses the results. Sect. 6 compares the dataset with other existing altimetry datasets as well as with TG data. The article closes with a conclusion and information about data availability.

## 2 Study area and input data

### 2.1 The North Sea

The North Sea is a semi-enclosed marginal sea of the North Atlantic Ocean situated on the north-west European shelf (Quante et al., 2016). It covers an area of about 570.000 square kilometers (970 by 580 km). It opens to the Atlantic through the Norwegian Sea in the north and has a second much smaller connection through the English Channel in the south. Moreover, it is connected to the Baltic Sea in the east. Its mean depth is around 94 m with shallow areas of less than 10 m depth in the southern part and much deeper parts (up to 700 m) in the Norwegian Trench area and parts of the Skagerrak. The average temperature is varying between about 5°C in February and 15°C in August (Huthnance, 1991). Sea level dynamics in the North Sea are driven by various forcings, namely tides (mainly semi-diurnal with a tidal range of up to 8 m), wind and atmospheric pressure, heat and water exchanges as well as river runoff and forcing from open boundaries (Zhang et al., 2020). The general circulation pattern in the North Sea is mainly cyclonic. The water flows southwards along the coastal areas of the British Isles, continues eastwards and northwards along the coasts, and finally leaves the basin as the Norwegian Coastal Current (Winther and Johannessen, 2006).





This study makes use of observation data collected in an area between 4°W and 12.2°E longitude and 50°N and 61°N latitude, except the regions of the Irish Sea and the Baltic Sea (Kattegat).

## 2.2 Satellite Altimetry data

Satellite altimetry has been providing sea surface height information since the launch of the Seasat mission in 1978. But only since 1992 at least two simultaneously measuring satellites are in orbit, ensuring more reliable and precise height observations as well as improved data coverage and temporal resolution. This study includes almost all available missions since ERS-2 in 1995, namely TOPEX (TP), ERS-2, Jason-1 (J-1), Envisat, Jason-2 (J-2), CryoSat-2 (CS-2), SARAL, Jason-3 (J-3), and Sentinel-3A/B (S3-A/B) (ordered by its launch dates). Data from the first phase of TP as well as from ERS-1 is not used because of doubts concerning their trend stability (Mitchum, 2000; Kleinherenbrink et al., 2019). NorthSEAL comprises the period between May 1995 and May 2019 (c.f. Fig. 1) and makes use of the latest data for all missions, updated by consistent external geophysical model corrections and ITRF2014-based orbits whenever available; see Sect. 3.1 for more details.

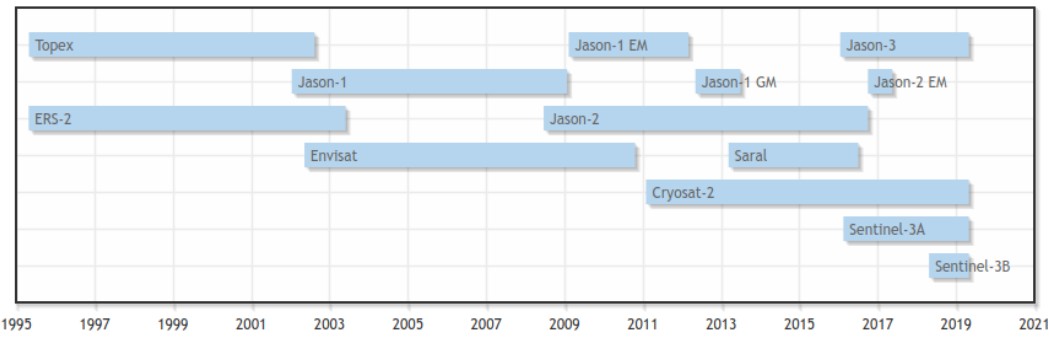

**Figure 1.** Satellite altimetry missions used in this study.

## 2.3 Tide gauge data

Monthy mean water level measurements of tide gauges are used for comparison and validation. The data is derived from the datum controlled database of the Permanent Service of the Mean Sea Level PSMSL (Holgate et al., 2013) as well as from the German Wasserstraßen- und Schifffahrtsverwaltung des Bundes (WSV, 2013). The WSV provides TG data for the German Bight with measurements at semi-diurnal tidal maxima and minima, i.e. at about 6-h resolution, which are first smoothed with a 2-day running mean filter and then down-sampled to monthly-mean observations. Values are retained which pass a 3-$\sigma$ outlier test.

Among all available stations in the North Sea region, those are selected that contain at least 80% valid measurements during the study period (1995 - 2019), resulting in a total number of 54 stations. The same monthly-averaged DAC-correction is applied as used for the altimetry data (Carrère and Lyard, 2003). To match the DAC-correction with the tide gauge records, among the nine closest points of the DAC grid the one that results in the highest variance reduction is selected. TG data is not





corrected for ocean tides, which are assumed to be removed by monthly averaging. Remaining influences from long-period tidal effects are assumed to have a smaller impact on the estimated trends than errors of ocean tide models would have directly at the coast.

## 2.4 Vertical Land Motion

In order to make trends determined from TG data comparable to absolute (i.e. geocentric (Gregory et al., 2019); with respect
to an ellipsoid) sea level trends from satellite altimetry, the relative TG measurements need to be corrected for vertical land motion (VLM).

VLM can be estimated from point measurements (e.g. from Global Positioning System (GPS) observations) or from regional or global models. Since these estimates differ significantly, data from different sources are used in this study. Beside GPS data and Glacial Isostatic Adjustment (GIA) models, also the non-linear effect of contemporary mass redistribution (CMR) on VLM
is applied.

GPS trends are derived from the dataset of the Nevada Geodetic Laboratory (NGL) (Blewitt et al., 2016), which contains more than 17.000 vertical velocities (IGS08 reference frame). Only GPS VLM estimates from measurements with a minimum record length of five years between 1995 and 2019 are considered. To combine the VLM estimates with the TG trends, the nearest GPS station within a radius of 50 km around a TG is used.

GIA VLM estimates are taken from two different models. The first one, ICE-6G D (VM5a) (Peltier et al., 2018) was refined by geodetic constraints primarily by GPS observations from the Jet Propulsion Laboratory (Desai et al., 2009) over 1994-2012 and from other complementary observations, such as Very Long Baseline Interferometry (VLBI) or Doppler Orbitography and Radiopositioning Integrated by Satellite (DORIS) (Peltier et al., 2015). The model-fit to those observations was improved by modifications of the glaciation history. The second GIA VLM estimate is from Caron et al. (2018) (hereinafter C18). The
solution is based on an ensemble of 128,000 model runs. Among those, the highest likelihood of parameters describing the ice history and 1-D Earth structure was identified from an inversion of GPS and relative sea level data using Bayesian statistics. The GIA estimate represents the expectation of the most likely GIA signal of the ensemble, and formal uncertainty estimates are directly inferred from the Bayesian statistics.

Ongoing changes in terrestrial water storage as well as mass changes in glaciers and ice sheets cause elastic responses of the
Earth which can result in nonlinear vertical movements. These effects from CMR are not captured by GIA models and only partially detected by GPS observations due to the relative shortness of the record lengths. Using GRACE satellite gravimetry observations, Frederikse et al. (2019) showed that associated time-varying solid Earth deformations can lead to very different trends (in the order of mm/year), depending on the time period considered during the last two decades. Therefore, VLM estimates from GIA are supplemented with CMR-related land motions as used and distributed by Frederikse et al. (2020). This
estimate is based on a blend of GRACE and GRACE-FO observations during 2003-2018, as well as process model estimates, observations and reconstructions for the period 1900-2003 (refer to Frederikse et al. (2020) for a detailed description of the datasets and sources used).

Figure 2 gives an overview on the applied VLM corrections. There are some noticable differences between the two GIA estimates. In particular, the dipolar feature spanning from Scotland to the German bight in the ICE-6GD solution (Figure 2a) is

much less pronounced in C18's estimate (Figure 2b). The linear CMR signal, which is computed over 1995-2019 (Figure 2d), shows a non-negligible uplift signal of about 0.7 mm/year in the entire region of the North Sea.

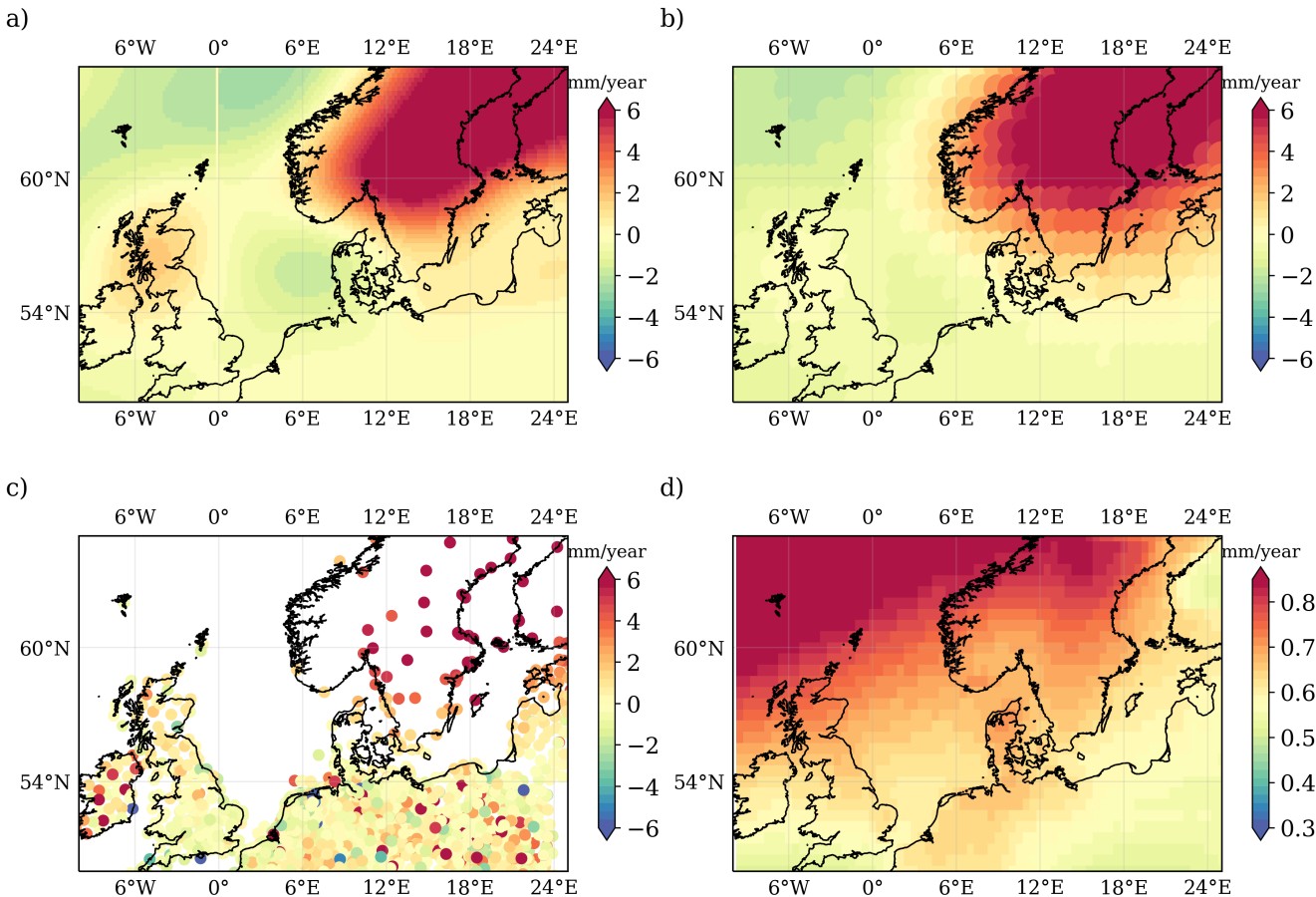

**Figure 2.** VLM estimates used to correct relative sea level trends from TG. GIA trends from a) ICE-6G D (VM5a) and b) Caron et al. (2018); c) GPS trend from the NGL solution; d) trend caused by contemporary mass redistribution over the period 1995-2019. Note that the scale of d) is much smaller than in the other plots.

## 2.5 External satellite altimetry products

For validation purposes, NorthSEAL is compared with other gridded altimetry products available for the region. These products are provided by CMEMS (Taburet et al., 2019) and SL_cci (Legeais et al., 2018). Both use a regular grid and have a coarser





spatial resolution of 0.25°. Given that the SL_cci dataset only lasts until 2015 (at the time of writing), all the following comparisons described in section 3.4 are only performed over the period May 1995-December 2015.

## 3  Methods

Most of the methods applied in this study have been developed in the frame of the European Space Agency's Baltic+ Sea Level (ESA Baltic+ SEAL) project. Thus, detailed information can be found in the Algorithm Theoretical Baseline Document
(ATBD) of that project (Passaro et al., 2020), in Passaro et al. (in review), and on http//balticseal.eu.

### 3.1  Along-track data preprocessing

In order to generate monthly sea level anomaly (SLA) grids, the altimetry along-track observations go through a chain of several preprocessing steps, including a retracking for conventional altimeters by the ALES retracker (Passaro et al., 2014) and for SAR altimetry by the ALES+SAR retracker (Passaro et al., 2020), an empirical adaption of the physical ALES+ retracker
(Passaro et al., 2018a) to SAR waveforms. Moreover, a relative multi-mission cross-calibration (Bosch et al., 2014) referencing all used altimetry missions to the TOPEX (and later to the Jason) data is performed. Sea level anomalies are computed using the following equations:

$$SSH = H_{orbit} - (R + WT + DT + IONO + OT + SSB + DAC + SET + PT + RC) \qquad (1)$$

$$SLA = SSH - MSSH \qquad (2)$$

Here, $H_{orbit}$, $R$ and $MSSH$ mean the orbital height of the satellite referred to the TOPEX/Poseidon ellipsoid, the altimeter range and the mean sea surface for reducing Sea Surface Heights (SSH) to SLA. In this study, the mean sea surface DTU15MSS (Andersen et al., 2016) is used, which includes 20 years of data from the period 1992-2012. The other terms in Eq. (1) describe several geophysical and atmospheric effects, which are considered for SSH computation. They are listed in Table 1. The sources
of the orbital heights $H_{orbit}$ are provided in the appendix Table A1.

In a post-processing step, the along-track sea surface heights are cleaned from possible outliers by applying the following four steps:

- Distance to coast: Elimination of observations closer than 3 km to the coast (TOPEX: 5 km)

- Retracking flag: Elimination of corrupt observations, flagged based on the quality of waveform fitting (retrack indicator
$\geq$ 0.3 for conventional altimetry and $\geq$ 0.1 for SAR altimetry)

- SLA threshold: Elimination of sea level anomalies exceeding the interval $\pm 1.5 m$

- Contextual along-track outlier search: Elimination of observations exceeding three-times the mean absolute deviation (MAD) from the local median (determined from a moving median with a kernel size of 1 second)

**Table 1.** Geophysical corrections applied to the along-track altimetry data

| Correction | Model | Reference | Missions |
| --- | --- | --- | --- |
| Dry Troposphere (DT) | VMF3 | Landskron and Böhm (2018) | all |
| Wet Troposphere (WT) | VMF3 | Landskron and Böhm (2018) | S3-A/B |
| | GPD+ | Fernandes and Lázaro (2016) | CS-2 |
| | GPD | Fernandes et al. (2015) | all others |
| Ionosphere (IONO) | NIC09 | Scharroo and Smith (2010) | all |
| Ocean Tide (OT) | FES2014 (ocean+loading) | Lyard et al. (2020) | all |
| Dynamic Atmospheric Correction (DAC) | DAC (IB+MOG2D) | CLS (2021) | all |
| Solid Earth Tide (SET) | IERS Conventions 2010 | Petit and Luzum (2010) | all |
| Pole Tide (PT) | IERS Conventions 2010 | Petit and Luzum (2010) | all |
| Sea State Bias (SSB) | MGDR | Gaspar et al. (1994) | TP |
| | ALES+SAR SSB | Passaro et al. (2020) | CS-2, S3-A/B |
| | ALES SSB | Passaro et al. (2018b) | all others |
| Radial Correction (RC) | MMXO18 | Bosch et al. (2014) | all |

More details on this flagging are provided by Passaro et al. (2020).

## 3.2 Gridding

All observations passing the outlier elimination are introduced into a least-squares gridding procedure (e.g. Koch (1999)). They are interpolated onto a triangular mesh (geodesic polyhedron) in order to generate monthly grids with a spatial resolution between 6 and 8 km. The gridding procedure mainly follows the process flow introduced by Passaro et al. (2020) and Passaro et al. (in review). It is therefore only briefly described in the following text passages. Mean SLA per grid node are computed by fitting an inclined plane ($h$) to the along-track observations:

$$h(x,y) = c_0 + c_1 x + c_2 y \tag{3}$$

A local Cartesian coordinate system $(x,y)$ is defined around each grid node, which represents the origin. The grid node height is provided by the coefficient $c_0$, and $c_{1,2}$ are the slope coefficients. They are not used for the following processing. All along-track observations within a radius, the so-called cap-size, of 150 km around each grid node are considered. They are spatially averaged, whereby a Gaussian weighting in consideration of their distance to the grid node is applied. The minimum weight at the cap-size edge is set to circa 10%. Furthermore, uncertainty information is added to the least-squares approach based on the MAD of sea level anomalies per mission and month within a sub-area (0° to 6°E and 54°N to 58°N). The chosen area is free of topographic features. The MAD provides a rough estimate of the SLA noise level of a certain mission in a certain month. More information is available in Passaro et al. (2020).





Within the gridding procedure, the observation are filtered again in order to reject outliers from the estimation of the coeffi-
cients. This is done by performing a three-stage outlier detection.

1. Application of a standard 3-sigma criterion to sea level anomalies within the cap-size

2. Iterative outlier detection based on estimated residuals by applying a standard 3-sigma criterion. The iterative outlier
   search stops, if no outliers are detected.

3. Application of a one-sided T-Test by testing standardised residuals against quantiles of the Student distribution (e.g. Koch
       (1999)). Observations that exceed the boundary limit at a 99th percentile level of the Student distribution are excluded
       from the final coefficient estimation.

The monthly grids undergo a final check by removing sea level anomalies that exceed a threshold of $\pm 2\ m$. For each grid
node, the monthly mean SLA are provided together with an estimate of the uncertainty. In addition, a quality flag indicates if
the node can be safely used or should be handled with care. This flag is allocated according to the standard deviation per node.
If the standard deviation exceeds a specified threshold and the node has less than 280 months of valid data, it is labelled as
'bad' quality (flag==1). The threshold is set to the 90th percentile of all SLA standard deviations averaged over time.

### 3.3 Estimation of trend and annual amplitudes

The monthly SLA grids provide the basis for estimating a sea level trend per grid node as well as the amplitude and phase of the
annual cycle. While a linear trend is fitted to the data, the annual cycle amplitudes are obtained from half of the differences of
the months with the maximum/minimum multi-year monthly means. The uncertainties are based on the combined uncertainties
of these months.

Trend uncertainties are derived while accounting for autocorrelated errors in the data using Maximum Likelihood Estimation
(MLE). To identify the most appropriate noise model, required to accurately estimate the trend uncertainties, we investigate the
fit of a variety of different stochastic noise model combinations as done in e.g. Royston et al. (2018). These are an autoregressive
AR(1) noise model, a power law plus white noise, a generalized Gauss Markov (GGM) plus white noise, a Flicker noise
plus white noise and an auto-regressive fractionally-integrated moving-average (ARFIMA(1,d,0)) model. For the considered
domain, we find that on average (of all altimetry observations in the North Sea) the AR(1) has the lowest mean (or median)
values of the Aikaike Information Criterion (AIC, Akaike (1998)) and the Bayesian Information Criterion (BIC, Schwarz
(1978)). Thus, this model is selected to assess formal parameter uncertainties. Uncertainties of the linear trend and the annual
amplitude are given at the 95% confidence interval.

For a more detailed description of the parameter estimation please refer to Passaro et al. (in review, 2020).

### 3.4 Comparison of the data with tide gauges

To evaluate the performance of NorthSEAL, we compare absolute sea level trends and annual amplitudes derived from satellite
altimetry and TG. Moreover, correlations between both time series at TG locations are analysed. To match the altimetry





sea level data with the TG measurements, we follow the approach of Oelsmann et al. (2021) which only uses the most highly correlated data in the comparison instead of taking the altimetry observation closest to the TG. Here, 20% of the best correlated gridded altimetry data within a radius of 200 km around a TG are selected. This region is hereinafter called the Zone of Influence (ZOI). Time series of sea level are computed from spatial averages of the altimetry data within the ZOI. Absolute sea level

trend deviations between altimetry and TG are subsequently derived by subtracting the TG measurements from the averaged altimetry data, whereby the correction for VLM is applied.

The uncertainties of absolute TG sea level trends $u$ are based on the combined uncertainties of the TG and the VLM trends ($u = \sqrt{u_{TG}^2 + u_{VLM}^2}$). In order to compute uncertainties of trend differences (for significance tests), the altimetry uncertainties are also taken into account. Differences (and their uncertainties) in the annual amplitudes are computed by subtracting

the amplitudes from the individual altimetry and TG time series and by computing the combined uncertainty of the annual amplitudes.

We note that the approach of using the ZOI significantly improves the comparability of sea level trends. Accordingly, the trend differences presented in Sect. 6 are on average about 20% smaller when using the ZOI approach instead of taking the closest altimetry grid point.

In contrast to trends and the annual cycle, correlations are computed at the closest valid (i.e., with at least 280 unflagged months of data available) altimetry node within a radius of 150 km. Correlations are derived from the detrended altimetry and TG time series. We apply the quality flag to the SLA dataset before computing correlations. This reduces the number of TG-altimetry pairs to 52. We use the same number of TG (52) also for the comparison with other altimetry products.

## 4   The NorthSEAL dataset

All data is stored in NetCDF format and span a time period from May 1995 to May 2019. It is provided on an unstructured triangular mesh, characterized by nearly equally-spaced grid node distances ranging from 6 to 8 km (geodesic polyhedron), for the entire region of the North Sea between 4°W and 12.2°E and between 50°N and 61°N with the exception of the Irish Sea and the Kattegat.

The SLA grids are provided in monthly resolution. Data gaps due to missing observations or gridded SLA exceeding $\pm 2\ m$

are set to undefined. File names (i.e. YYYY_MM.nc) indicate year (YYYY) and month (MM). All provided coordinates and height values are referenced to the TOPEX ellipsoid. SLA data are referenced to the DTU15MSS (Andersen et al., 2016). The file named NorthSea_trend_and_annual_cycle.nc contains the sea level trends and amplitudes of the annual cycle per grid node. Table 2 lists all NetCDF variables included in the dataset.





**Table 2.** NetCDF variables included in the dataset

| NetCDF variable | description | unit |
|---|---|---|
| Monthly SLA grids | **YYYY_MM.nc** | |
| lon | Geographic longitude of grid node | degree |
| lat | Geographic latitude of grid node | degree |
| time | Day since 1985-01-01 00:00:00 (continuous) | day |
| sla | Sea Level Anomaly (SLA) | meter |
| sla_std_lsq | Uncertainty of SLA per grid node resulting from the gridding procedure | meter |
| num_used_obs | Number of observations used within the gridding procedure per grid node | - |
| num_obs | Number of theoretically available observations for gridding procedure per grid node | - |
| qf_monthly_grid | Quality flag resulting from the gridding procedure (Bad=1, Good=0) | - |
| mss | Mean sea surface height from DTU15MSS (Andersen et al., 2016) | meter |
| Trend grid | **NorthSea_trend_and_annual_cycle.nc** | |
| lon | Geographic longitude of grid node | degree |
| lat | Geographic latitude of grid node | degree |
| sla_trend | Sea level trend over May 1995 until May 2019 | meter/year |
| sla_trend_unc | Uncertainty of sea level trend (95% confidence level) | meter/year |
| sla_annual_ampl | Annual amplitude over May 1995 to May 2019 | meter |
| sla_annual_ampl_unc | Uncertainty of annual amplitude | meter |
| sla_annual_ampl_max | Month with maximum amplitude of the annual cycle | - |
| sla_annual_ampl_min | Month with minimum amplitude of the annual cycle | - |
| qf_monthly_grid | Quality flag resulting from gridding procedure (Bad=1, Good=0) | - |

## 5 Results

### 5.1 Sea Level Anomalies

Figure 3 shows the mean SLA, averaged over the observation period of 24 years. In addition, three selected time series at different locations in the North Sea are displayed. As described in Sect. 3.1, the SLA are referenced to the DTU15 mean sea surface. Since the input data and time period of DTU15MSS (1992-2012) and NorthSEAL (1995-2019) are different, the SLA do not average to zero everywhere. A geographical pattern of negative offsets around 2 cm is visible in large parts of the coastal areas of the North Sea, especially along southern Norway and the coasts in the south. This effect is probably due to the use of a global MSS product that uses a different set of retracked data and geophysical corrections. It has no influence on the time series analysis of SLA, in particular not on derived trends and annual amplitudes derived in this study. However, the pattern

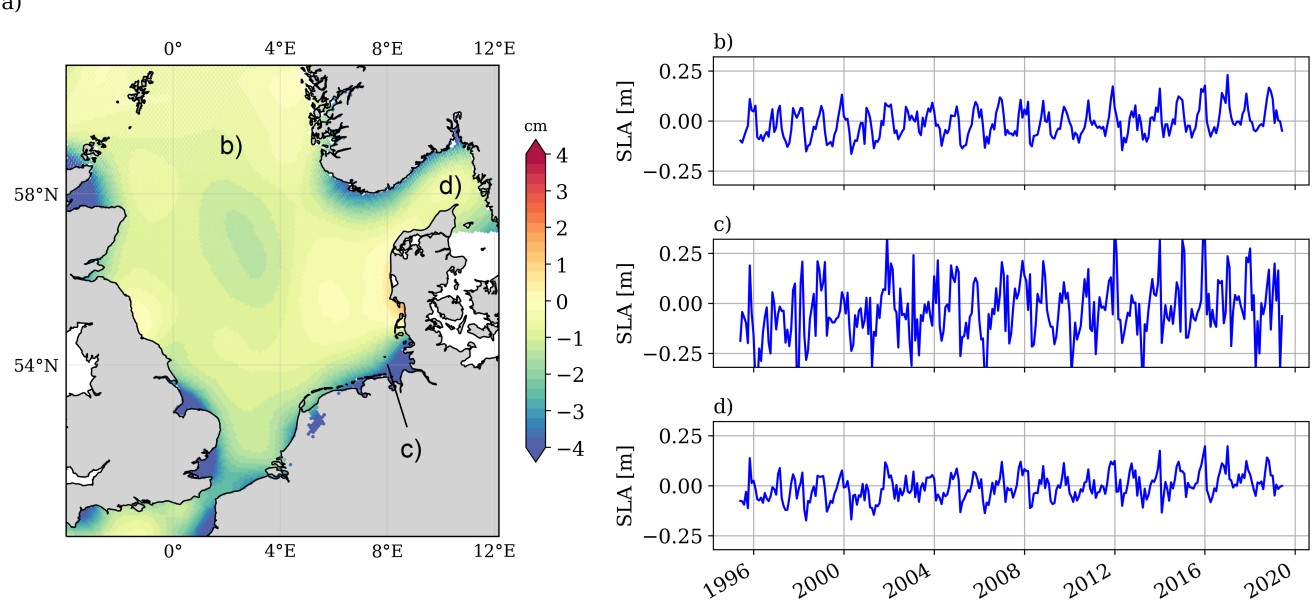

**Figure 3.** Mean sea level anomalies over the period May 1995 to May 2019 in the North Sea (a) and three examples for time series of sea surface anomalies at different locations (b-d)

should be kept in mind when using the SLA dataset for spatial studies. In fact, most of the affected regions are edited when taking the quality flag (c.f. Sect. 3.2) into account.

The three SLA time series on the right hand side of Fig. 3 show the temporal evolution of sea level for different grid nodes. A distinct annual oscillation as well as inter-annual changes are clearly visible for the three locations. Moreover, all three curves show a rise of sea level.

The monthly SLA time series displayed in Fig.3b-d suggest that the sea level variability differs strongly over the domain. In agreement with e.g. Dangendorf et al. (2014) and Wahl et al. (2013), the variance of the time series towards the German
Bight (Fig. 3c) strongly exceeds the variance observed at the Norwegian and the British coastlines (Fig. 3b and d). Using long TG records, Dangendorf et al. (2014) demonstrated that the sea level variability at the southern/eastern coastlines of the North Sea is particularly dominated by westerly winds, which explains up to 80% of the observed variability in these regions. The regional differences in variability also influence the estimated sea level trends and their associated uncertainties as with a length of 24 years the time span is still relatively short.

**5.2    Sea Level Trends**

Besides the mean sea surface, the temporal evolution of sea level is of great interest. In view of global change, especially the long-term change is highly relevant for predicting future sea levels and their impact for society and environment. Thus, as part of NorthSEAL, sea level trends are also provided. While the mean sea level trend between May 1995 and May 2019 in the



North Sea amounts to 2.61 ± 0.95 mm/year, the trend varies between 1.5 and 3.5 mm/year over the region as illustrated in Fig. 4. Highest trends are observed in the German Bight and around Denmark, whereas significantly lower trends are observed around the southern part of Great Britain. Figures 4a) and b) include a black contour line. This line confines the coastal areas flagged in the SLA dataset in the frame of the gridding procedure (cf. Sect. 3.2). This flag is defined quite conservatively and tuned to provide optimal SLA time series. For trend computations, a rejection of the flagged areas is not necessary when the trend uncertainties are taken into account in the course of the data analysis. When the flagged coastal regions are excluded, the overall trend for the North Sea changes only marginally to 2.60 ± 0.95 mm/year.

Even with almost two and a half decades of observation data, the trend uncertainties are still in the same order of magnitude as the trend itself. As visible from Fig. 4b), the trend uncertainties vary between 0.5 and 2.5 mm/year with smallest values in the northern part of the North Sea, especially close to the Norwegian coast. The highest uncertainties can be seen in the German Bight and at some smaller bays along the coasts of Great Britain and France. The particularly large uncertainties in the German Bight coincide with the aforementioned larger sea level variability described by Dangendorf et al. (2014) and may also be affected by poorer ocean tide corrections in these areas.

The average trend of 2.61 ± 0.95 mm/year differs from the MSL trend of 4.00 ± 1.53 mm/year observed by Wahl et al. (2013) over 1993-2009. The difference is, however, still within the limits of the confidence bounds. Deviations between the trends may on the one hand be due to the different periods of observation. On the other hand, the trend of Wahl et al. (2013) is based on TG observations and may thus not unequivocally be compared to the sea level trend from gridded altimetry data that is spatially distributed over the North Sea. Compared to the sea level trend of the North Sea over the last century of 1.53 ± 0.16 mm (Wahl et al., 2013), our study reveals a significantly increased sea level trend over the past two and a half decades. Again, however, the value from (Wahl et al., 2013) is based on TG and not entirely comparable. In general, our observed average trend is well in the order of the global sea level trend of 3.1 ± 0.1 mm/year (from 1995 to 2018) reported by Cazenave et al. (2018).

## 5.3 Annual Amplitudes

In addition to long-term changes in sea level, NorthSEAL allows for the analysis of seasonal variations. Figure 5 shows the estimated amplitudes of the annual cycle as derived from multi-year monthly means (see Sect. 3.3). Highest amplitudes of more than 10 cm are visible in the German Bight and close to the Danish coasts. The signal is much smaller in the north of the region around Norway. Estimates of the annual amplitudes are much more accurate than the estimates of the trends. Uncertainties vary between 1 and 4 cm and amount to approximately one third of the signal itself.

## 6 Comparison with tide gauge measurements and external altimetry datasets

Detrended SLA time series of NorthSEAL are compared with measurements from TG stations and the alternative altimetry products from CMEMS and SL_cci (see Sect. 2.5) using the methods described in Sect. 3.4. In order to be as consistent as possible, the analysis is performed for the overlapping time period of all three altimetry datasets (1995-2015).

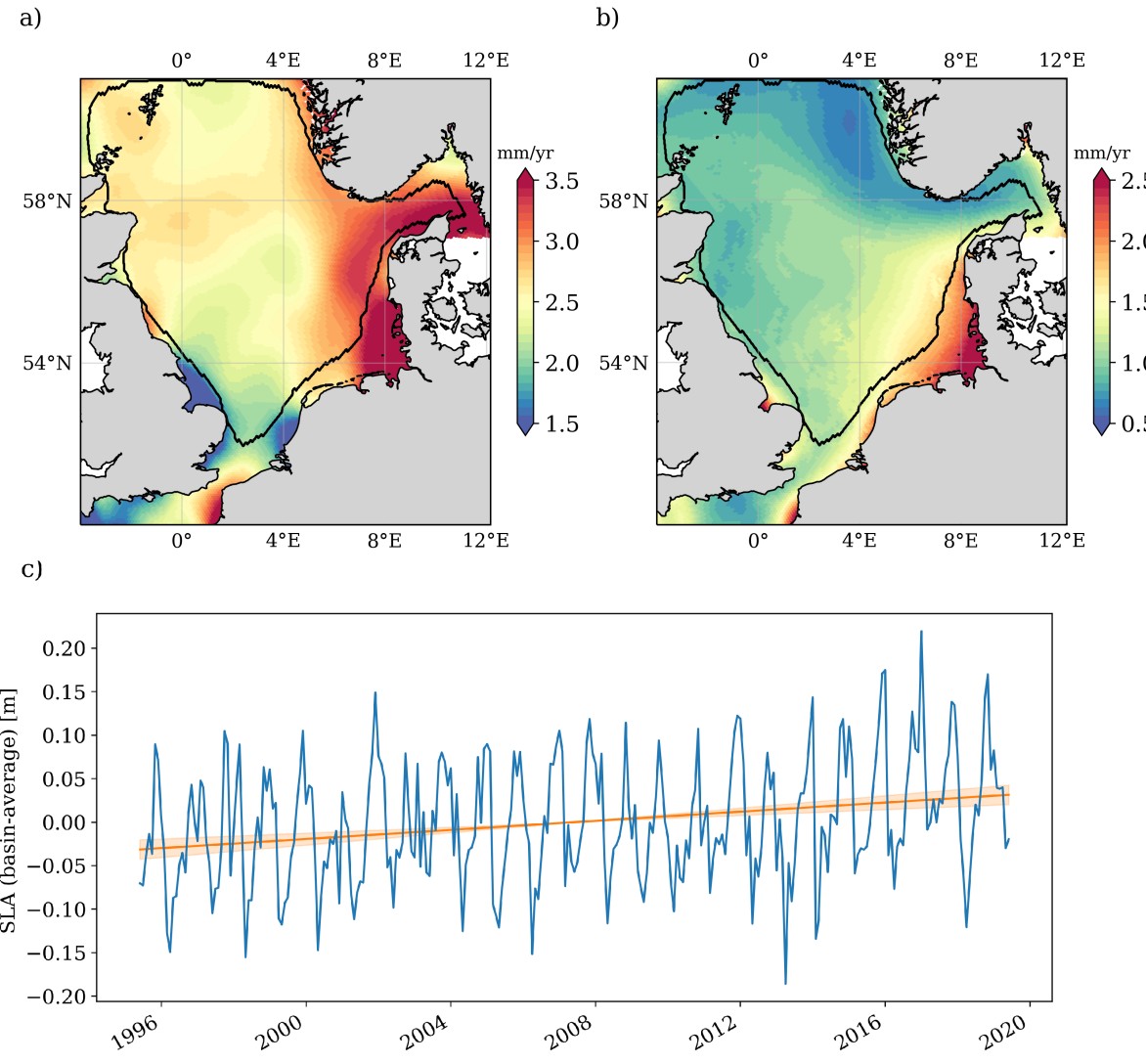

**Figure 4.** Sea level trends from altimetry (a) and their associated uncertainties (b) for the period between May 1995 and May 2019. The black contour line confines the coastal regions flagged in the dataset. The curve in (c) shows the SLA time series averaged over the entire North Sea. The linear trend is $2.61 \pm 0.95$ mm/year.





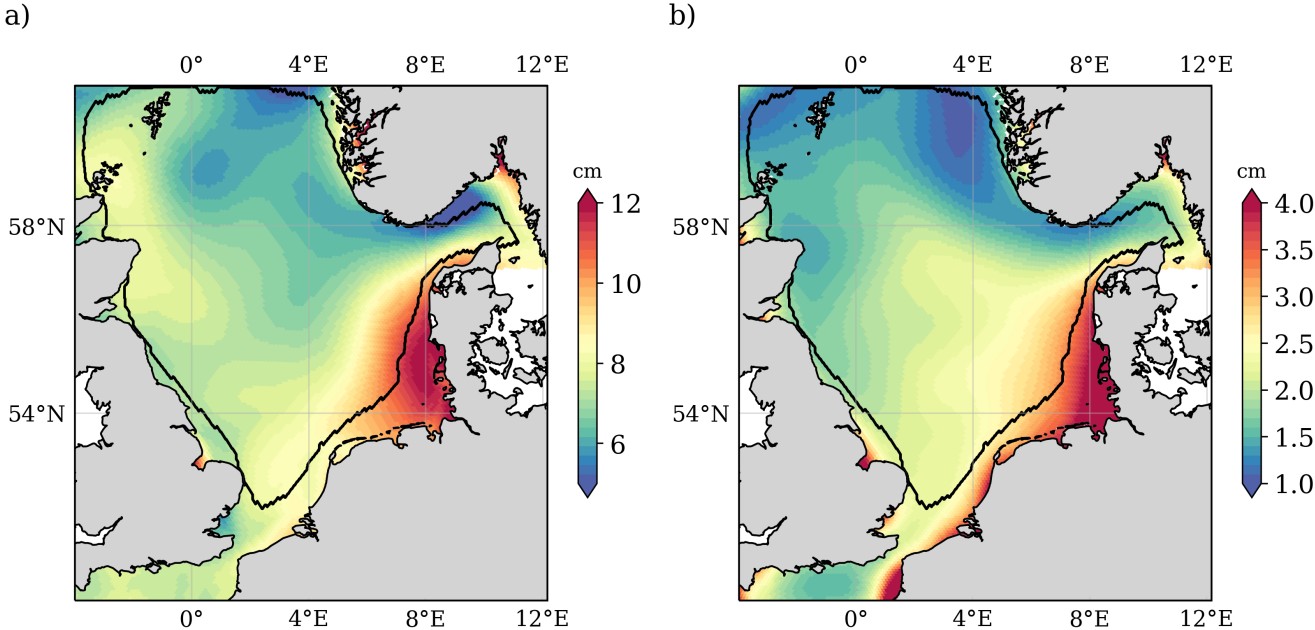

**Figure 5.** Amplitudes of the annual cycle of the sea level (a) and associated uncertainties (b) for the period 1995-2019. The black contour line confines the coastal regions flagged in the dataset.

## 6.1 Correlation

The median and mean correlation between NorthSEAL SLA and 52 TG amounts to 0.86 and 0.85 respectively (Fig. 6a.) Highest correlations (up to 0.93) are found for the TG at the Shetland Islands as well as along the northern coast of Denmark. Lowest correlations appear in small bays and fjords, e.g. the Firth of Forth (0.69) and the Oslofjord (0.72). Low correlations are also observed along the coast of Belgium and the Netherlands at the mouth of the English Channel (Dover strait).

This distribution is also visible in the correlations between the CMEMS/SL_cci products and TG (Fig. 6b+c). Very likely, some of those TG in river mouths or smaller bays do not provide data that is representative for the sea level variations of the coastal or open ocean in their vicinity. Dividing all TG stations into different categories according to their locations (Fig. 7) demonstrates that the majority of optimally located stations show correlations of around 0.8 and better, whereas stations located at fjords, rivers or close to flood gates show generally lower correlations. For stations located in river mouths the spread of correlation values is largest. On the other hand, stations sheltered by islands show high correlations with a small spread. For TG at rivers and near flood gates NorthSEAL clearly outperforms the other two products. Obviously, here, the quality flag prevents the use of inappropriate data successfully. However, in Fjords, especially in the Oslofjord, NorthSEAL shows lower performance. This may be caused by a too large distance between the respective grid node and the TG.

Overall, with median and mean correlations of 0.86 and 0.85 respectively, NorthSEAL matches the TG measurements better than SL_cci (0.82/0.78) and CMEMS (0.79/0.78). Only for few TG, the correlations are lower than for SL_cci (23.1%)

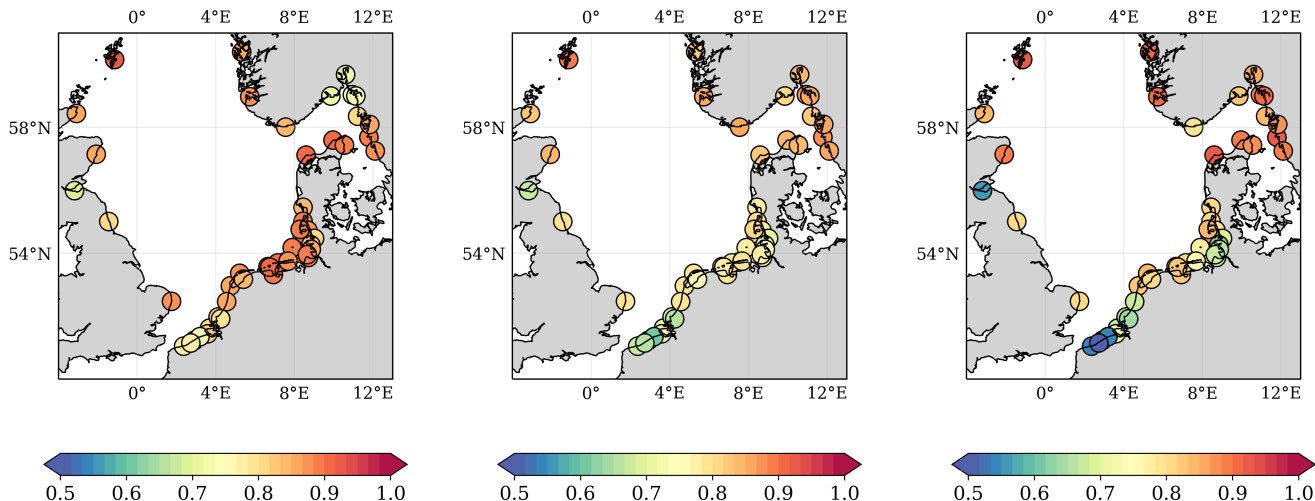

**Figure 6.** Correlations between altimetry sea level anomalies and measurements of 52 TG, computed over a period of 20 years (1995-2015) for three different altimetry datasets: a) NorthSEAL b) CMEMS c) SL_cci. The median correlation for all stations is 0.865 (NorthSEAL), 0.789 (CMEMS), and 0.816 (SL_cci). Quality flags of NorthSEAL grids have been considered.

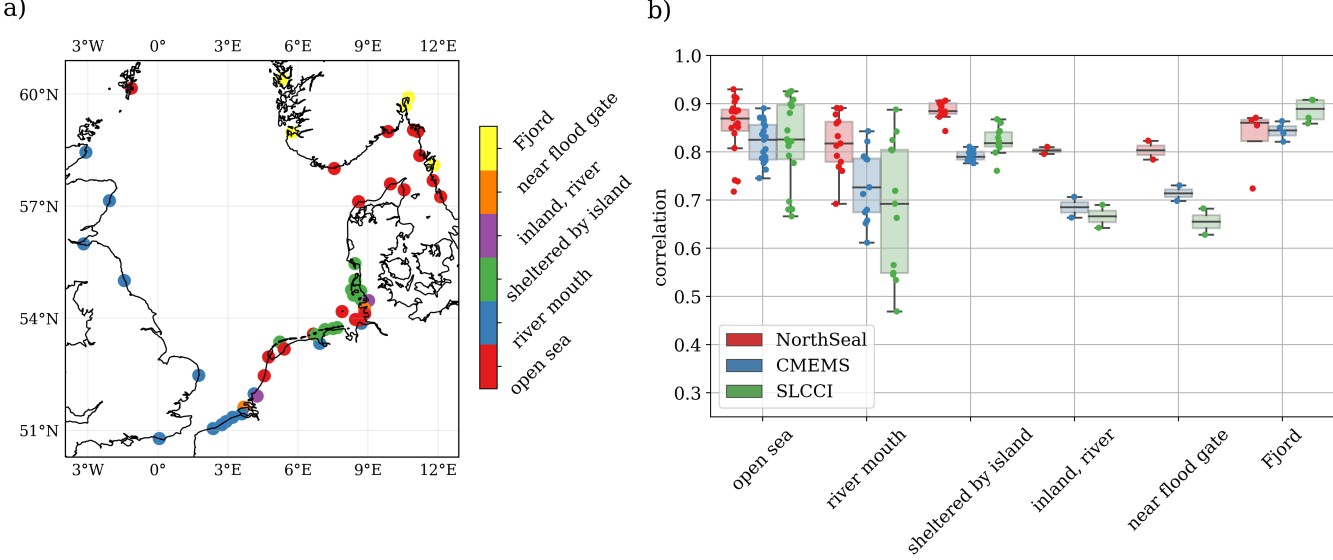

**Figure 7.** Correlations between sea level variations from three different altimetry products and TG measurements depending on TG location category.





and CMEMS (11.5%). The average difference in correlation is about 0.06 for both datasets (0.0651 for SL_cci; 0.0622 for CMEMS). This indicates an average improvement in correlation between 8.4% (CMEMS) and 10.5% (SL_cci).

The better performance close to the coast can be attributed to a large part to the consideration of quality flags in the North-
SEAL SLA grids. They enable the selection of the closest grid node with reliable SLA information and improve the correlations with TG measurements by about 12% on average. Moreover, the flag led to the exclusion of two TG stations for which no valid grid node could be found within a distance of 150 km (Oslo in Norway and Newhaven at the south coast of Great Britain). Figure A1a) in the appendix shows the correlations between altimetry and TG if the quality flags are not considered, i.e. if the SLA series at the grid node closest to the TG is applied.

## 6.2   Differences in trends and annual amplitudes

As described in Sect. 3.4, TG trends are corrected for VLM in order to make them comparable with the absolute sea level trends from satellite altimetry. Figure 8 shows the absolute sea level trends from three altimetry datasets and from TG measurements (top row), their standard deviations (middle row) and the trend differences at the TG locations (bottom row). Since GPS information is not available for all TG locations (see Sect. 2.4), the figure only contains 27 TG.

The trends from the three altimetry datasets show clear regional differences. While all products show higher trends around Denmark, discrepancies are visible in the central North Sea. For large areas south of 56°N, the trends from SL_cci and (to a smaller extent) CMEMS are about 1 mm/year higher than the NorthSEAL trends. Likewise, both products show higher trends close to the coasts of Denmark and Norway. These differences are up to about 1.5 mm/year. However, none of these differences is significant in view of the trend uncertainties.

The trend uncertainties show similar geographic patterns in all datasets. Lowest accuracies exist in the area of the German Bight. They improve towards the central North Sea and are highest in the region of the Shetland Islands and west of Norway. The main differences between the three datasets can be seen in small bays (e.g. along the British coast), where NorthSEAL is characterized by higher uncertainties than the two other products. Based on the existing data, it cannot be decided whether this behaviour is caused by less correct trends or by more realistic uncertainties.

The comparison of altimetry and TG trends (the latter corrected for VLM) reveals large discrepancies, while the uncertainties for both data types agree quite well. Trend differences reach up to 3.8 mm/year with a median of -0.13 mm/year and an RMS of 1.40 mm/year. Table 3 indicates that the RMS values for NorthSEAL are slightly smaller than for the two other altimetry datasets (CMEMS: 1.42 mm/year, SL_cci: 1.49 mm/year). However, since the time period of about 20 years is still quite short for a reliable trend estimation, and since uncertainties of three different data types are involved (altimetry, TG, and VLM), the
trend differences for almost all locations are statistically not significant.

Moreover, the values are dependent on the applied VLM correction of the relative TG trends. Table 3 provides the RMS of trend differences as well as the median bias between the trends from altimetry and TG. For consistency, these results only refer to the 27 TG that are co-located to a GPS station. Interestingly, implementing of the GIA VLM correction C18 (Caron et al., 2018) results in a lower deviation of the trends than using local GPS information. For NorthSEAL, for example, the
improvement is about 18% in terms of RMS. This VLM correction also outperforms the second GIA-based estimate from ICE-

**Figure 8.** Absolute sea level trends over the period 1995-2015 (a-c) and associated uncertainties (d-f) for the three altimetry datasets (North-SEAL, CMEMS, SL_cci) and TG measurements corrected for VLM. Differences in absolute sea level trends in TG locations (g-i). None of of these differences are statistically significant.

**Table 3.** Comparison of absolute sea level trends from altimetry (for NorthSEAL, CMEMS, SL_cci) and 27 TG (for which GPS data are available) over 1995-2015. Trend differences between altimetry and TG are provided in terms of root mean square (RMS) difference and median bias in mm/year. For each altimetry product, the best solution is marked in bold.

|  | NorthSEAL | | CMEMS | | SL_cci | |
|---|---|---|---|---|---|---|
|  | RMS | BIAS | RMS | BIAS | RMS | BIAS |
| NGL(GPS) | 1.4010 | -0.134 | 1.4248 | 0.145 | 1.4905 | 0.424 |
| C18 | **1.1461** | **0.038** | 1.1577 | 0.234 | 1.3061 | 0.509 |
| ICE-6G | 1.3131 | 0.181 | 1.4312 | **0.096** | 1.5711 | 0.484 |
| C18 + CMR | 1.2755 | -0.596 | **1.1514** | -0.400 | **1.1580** | **-0.072** |
| ICE-6G + CMR | 1.3219 | -0.421 | 1.3204 | -0.475 | 1.3466 | -0.099 |

6G. The better performance of GIA-based VLM corrections compared to GPS corrections could be caused by the relatively large maximum allowed distance of 50 km between TG and GPS station. Locally unequal VLM might introduce different signals at GPS and TG location even over such distances. For example, two GPS stations located on the island of Sylt (RANT and HOE2, in the vicinity of TG Hörnum) indicate VLM estimates differing by more than 1 mm/year, even though they are

within a distance of only about 10 km. This could make a smooth long term GIA model make a better suited correction. As can be seen in Fig. 8g)-i), the largest scatter of absolute sea level trends is observed in the German Bight, more precisely at the offshore-located islands. Such areas could be stronger affected by local VLM than, for instance, the TG locations along the British coastlines.

Adding the effect of CMR to GIA estimates influences the agreement of the trends in different ways. We observe the

strongest improvement for SL_cci (for both RMS and bias) and moderate improvement for the RMS of CMEMS, however with an increase in the bias. Likewise, the bias becomes larger for NorthSEAL when the effect of CMR is added, and in this case, also the RMS increases (for either GIA-solution). The applied CMR correction generates a large-scale uplift signal of about 0.7 mm/year over the domain (Fig. 2), which leads to an increased absolute sea level trend at the TG. This effect projects into the negative biases for NorthSEAL and CMEMS. The highest consistency between altimetry and TG is obtained for NorthSEAL,

when the TG measurements are corrected using C18 VLM. Nevertheless, further investigation are worthwhile to understand why the CMR correction introduces pronounced biases for NorthSEAL and CMEMS while it improves the consistency for SL_cci. For such studies, however, longer time series would be desirable in order to reduce the trend uncertainties.

A comparison of the annual cycles among the altimetry products and TG data reveals a much better consistency than in the case of sea level trends. Figure 9 shows the amplitudes (top row), their uncertainties (middle row) and the discrepancies

between altimetry and TG (bottom row). Overall, we find qualitatively similar patterns of the amplitudes as well as of their uncertainties.

The difference between altimetry and TG amplitude is significant for only very few stations (three stations for North-SEAL and SL_cci, seven stations for CMEMS). The largest deviation is found for CMEMS at Dagebüll in the German Bight



**Figure 9.** Amplitudes of the annual cycle (a-c) and associated uncertainties (d-f) for the three altimetry datasets and TG data. Differences of annual amplitudes at TG locations are shown in (g-i). None of of these differences are statistically significant.





(-6.0 cm). This value is just slightly larger than the combined uncertainty of 5.8 cm (95% confidence level). The German Bight
is generally characterized by the largest annual amplitudes and largest uncertainties. Note, that a negative deviation means
an underestimation of the amplitude in the altimetry dataset. For the whole domain, NorthSEAL shows the lowest absolute
mean deviations of the annual cycle of 1.3 cm (CMEMS: 1.8 cm; SL_cci: 1.5 cm). This consolidates its coastal performance
compared to the other altimetry products.

## 7  Conclusions

This paper presents the new dataset NorthSEAL of monthly gridded sea level heights in the North Sea over the period 1995-
2019. NorthSEAL contains sea level anomalies, long-term mean sea surface heights (DTU15MSS), uncertainty estimates as
well as quality flags on a triangular 6-8 km mesh. In addition, derived linear sea level trends and amplitudes of the annual
cycle are provided per grid node. An updated mean sea surface optimized for the use together with the sea level anomalies of
NorthSEAL is subject to future work.

NorthSEAL has been created from 24 years of multi-mission cross-calibrated altimetry data, carefully preprocessed and opti-
mized for coastal applications. Along-track data was gridded using the innovative procedure developed within the ESA Baltic+
SEAL project. Comparison with existing global altimetry products and with TG observations revealed an improved perfor-
mance with average correlations between SLA and TG time series of 0.85 and median trend differences of $0.04 \pm 1.15$ mm/year
(C18 VLM). Since the investigation period is relatively short, the uncertainties are still too high to see statistically significant
trend differences.

With NorthSEAL, for the first time, a regionally optimized sea level dataset for the entire North Sea is available. Even though
the length of the time series is still much shorter than some of the TG records in the region, the data offers the possibility to
investigate sea level changes – not only in coastal but also in offshore areas. This enables basin-wide studies of physical
processes driving sea level variability, such as the impact of atmospheric wind and pressure forcing. Moreover, NorthSEAL
can also contribute to improve sea level projections, and it provides an observational basis for planning coastal protection
measures. Examples include national and international dike building projects, or an improved understanding whether projects
like the Northern European Enclosure Dam NEED (Groeskamp and Kjellsson, 2020) can be a solution for mitigating sea-level
related climate change impacts.

## 8  Data availability

The NorthSEAL dataset (i.e. monthly sea level anomalies, sea level trends and annual amplitudes) can be downloaded from
SEANOE at https://doi.org/10.17882/79673 (Müller et al., 2021). The altimetry observations used and all necessary atmo-
spheric as well as geophysical corrections are obtained from the Open Altimeter Database (OpenADB) operated by DGFI-
TUM (https://openadb.dgfi.tum.de/en/, last access: 05 March 2021). Original altimeter datasets are maintained by AVISO,
ESA, NOAA, and PODAAC. GPS vertical velocity estimates are provided by the NGL at the University of Nevada (http:



//geodesy.unr.edu, last access: 1 September 2020 - Blewitt et al. (2016)). PSMSL tide gauge data are available at https://www.psmsl.org/data/obtaining/ (last access: 10 December 2020 - Holgate et al. (2013)). Additional German tide gauges were obtained from the Wasserstraßen- und Schifffahrtsverwaltung des Bundes (WSV) and are available on request through the Bundesanstalt für Gewässerkunde (BfG) (https://www.bafg.de,last access: 9 October 2020 - WSV (2013)). The CMEMS dataset (averaged DT-MSLA AVISO gridded altimetry data) are provided from AVISO (https://www.aviso.altimetry.fr, last

access: 10 December 2020). The SL_cci (Sea Level ECV v2.0) product is available at https://doi.org/10.5270/esa-sea_level_cci-IND_MSL_MERGED-1993_2015-v_2.0-201612 (last access: 27 July 2020 - Legeais et al. (2018)). The GIA dataset is available from JPL/NASA (https://vesl.jpl.nasa.gov/solid-earth/gia/, last access: 1 September 2020 - Caron et al. (2018)). The VLM is distributed at https://www.atmosp.physics.utoronto.ca/~peltier/data.php (last access: 25 November 2020 - Peltier et al. (2018)). The contemporary mass redistribution is provided at ZENODO (https://zenodo.org/record/3862995#.X05RrIuxVPY,

last access: 1 September 2020 - Frederikse et al. (2020)).

## Appendix A: Supplementary material

**Table A1.** Orbit solutions used for the different altimetry missions

| Mission | Orbit solution | Institution | Reference frame | More info |
| --- | --- | --- | --- | --- |
| TP | MGDR (EPN Type) | NASA/GSFC | old ITRF | AVISO (1996) |
| J-1 | VER13 | GFZ | ITRF2014 | Rudenko et al. (2018) |
| J-2[1] | std1504_14 | GSFC | ITRF2014 | Lemoine et al. (2015) |
| J-3 | GDR-F | CNES | ITRF2014 | CNES (2018) |
| ERS-2 | Reaper V2 | DEOS | ITRF2014 | Otten and Visser (2019) |
| Envisat | VER13 | GFZ | ITRF2014 | Rudenko et al. (2018) |
| SARAL | GDR-F | CNES | ITRF2014 | CNES (2018) |
| CS-2 | GDR-F | CNES | ITRF2014 | CNES (2018) |
| S3-A/B | GDR-F | CNES | ITRF2014 | CNES (2018) |



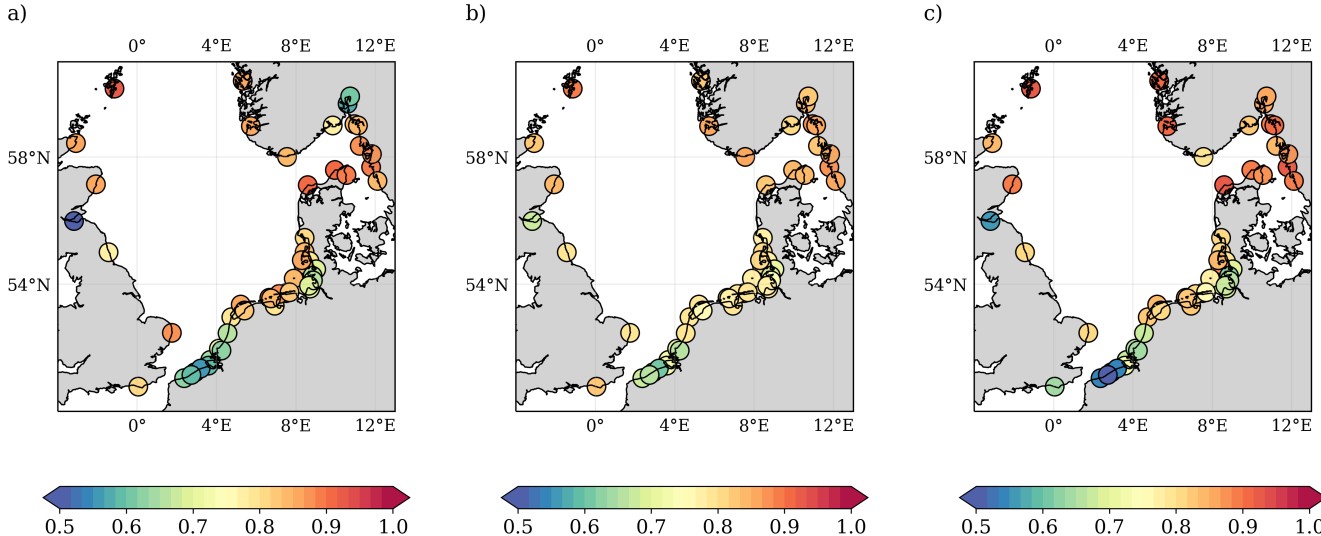

**Figure A1.** Correlations of altimetry sea level observations with 54 tide gauge observations, computed over a period of 20 years (1995-2015) for three different altimetry datasets: a) NorthSEAL b) CMEMS c) SL_cci. The median correlation for all stations is 0.828 (NorthSEAL), 0.790 (CMEMS), and 0.816 (SL_cci). Without taking into account quality flags of NorthSEAL SLA grids.

*Author contributions.* D.D. designed the study together with all co-authors. D.D., J.O., and F.L.M. wrote the manuscript. F.L.M. and J.O. were responsible for the computation of the data and prepared the NetCDF datasets. F.L.M. prepared the SLA grids and J.O. prepared TG and VLM data and performed the TG comparison. M.P. was responsible for the altimetry data retracking and the sea state bias correction, and he was principle investigator of the BalticSEAL project, in which the majority of the methods used were developed. C.S. is responsible for the altimetry database in DGFI-TUM and ensured the availability of the along-track observations used in this study. J.B. and M.R. were responsible for and reviewed all deliverables of the ESA Baltic+ SEAL (Sea Level) project enabling the method development and the generation of this dataset. F.S. initiated the study and provided the basic resources at DGFI-TUM making the study possible. All authors contributed to the discussion and interpretation of the results. In addition, they all read, commented and reviewed the final manuscript.

*Competing interests.* The authors declare that they have no conflict of interest.

*Acknowledgements.* Most of the methods used for the generation of the dataset have been developed in the framework of the Baltic+ SEAL (Sea Level) project (ESA Contract: 4000126590/19/I-BG), funded by ESA's Scientific Data Exploitation Element of the Earth Observation Envelope Programme (EOEP-5), Baltic Regional Initiative.

We highly appreciate the opportunity of using the extensive collection of noise models provided by the Hector software (Bos and Fernandes, 2019).





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
