# Peer review of "NorthSEAL: A new Dataset of Sea Level Changes in the North Sea from Satellite Altimetry"

_Earth System Science Data, 2021_

## Author Comment (AC1)

We thank the reviewer for the valuable feedback. Below, we provide an answer to each point and explain the corresponding changes adopted in the manuscript. The original review comments are copied from the report (in black) and our answers are in blue. Manuscript changes are in **bold italic**.

Anonymous Referee #1

Main comments

*Introduction*

L41-52: as part of their motivation for an improved altimetry dataset for the North Sea, the authors point out that most studies investigating MSL changes in the North Sea are based on TGs, which do not provide open ocean information. While it is true that TGs cannot provide open ocean information, multiple studies have used existing altimetry data and regional models to gain more understanding of the processes driving sea-level change & variability and its spatial characteristics in the North Sea and on the wider continental European shelf, for example: https://doi.org/10.1002/2014JC009901, https://doi.org/10.1002/2016GL070750, https://doi.org/10.1029/2020JC016325, https://doi.org/10.1007/s00382-020-05378-0, https://doi.org/10.1029/2003GL017041, https://doi.org/10.1029/2012JC008285. Including more of these and other relevant studies would make the introduction more comprehensive. The authors may also consider to draw the connection with modeling studies for the North Sea – one of the applications of (improved) altimetry data is model validation and this is currently not mentioned in the manuscript.

We agree with this comment. We will include an additional paragraph on ocean models in the introduction:

***An additional source for studying the sea level and its variation are physical ocean models. These have already been used in the North Sea, the European Shelf and the North Atlantic for many years (Flather, 2000; Wakelin et al., 2003). In contrast to observations, most models do have an improved resolution and a regular coverage. However, limitations may exist due to incomplete process descriptions or doubtful assumptions, especially when no data assimilation is implemented. Observations and observation-based datasets like North SEAL play an important role to validate and improve pure model simulations (Hermans et al., 2020; Tinker et al., 2020).***

Moreover, the conclusion will be updated to account for the application of the new data set for model validation and assimilation (see also one of your next comments).

In light of this I am also wondering whether the authors have considered extending their dataset to the whole northwestern European continental shelf?

In principle, the approach is applicable to all regions. Especially, it will be valuable in coastal areas with complicated coastlines and small islands and in sea-ice covered areas. Thus, the extension to the northern European coast would be just logical. So far we have no concrete plans for an extension, but we might consider this for the future.

***Altimetry comparison with TGs***

L71: *'i.e. trends and annual amplitudes'* the motivation for evaluating these metrics specifically is missing. The analysis of the correlation of monthly mean timeseries (6.1) could also be mentioned. Additionally, evaluating the interannual variability of annual mean sea level would also be very helpful for future studies.

Trends and annual amplitudes are listed here, because they are part of the published dataset (in addition to the monthly SLA grids). We consider these to be the most interesting parameters for most of the users. In fact, other parameters can also be computed based on the SLA grids, however, these are not part of the product and should be computed individually by the users.

We will add our motivation to provide these parameters in the manuscript: *... and estimated parameters, i.e. trends and annual amplitudes,* **which are considered to be of special interest for many users.**

However, we agree that the manuscript will benefit from adding additional metrics in the validation part. Based on your comment, we will add a RMSE analysis of SLA time series, containing also de-seasoned and annual values (see also answers to your following comments).

Section 3.4: should establish both how and why these metrics are compared, and in combination with Section 6.1 raises the question why only the correlation of detrended monthly means is assessed but not the magnitude of the difference?

Based on your comment, we will include an assessment of the magnitude of the SLA differences in the paper, namely an analysis of RMSE. This will not be limited to the de-trended time series in monthly resolution but also comprise annual time series as well as de-seasoned time series.

We will mention the RMSE at the beginning of Section 3.4 and will include an additional paragraph at the end of this section saying:

**Next to correlations, we also analyse the RMS difference of TG and altimetry time series, in order to study the spread between the data. To account for datum differences between TG and altimetry, offsets are removed from the difference time series. Three different solutions are computed: monthly (de-trended), monthly (de-trended and de-seasoned), as well as annual (de-trended). In order not to distort the annual values by incomplete years, only the period from January 1996 to December 2016 is used for all comparisons. To de-season the data, we subtract the multi-year monthly averages. Annual averages are studied to compare the variability on inter-annual time scales and to also assess the correctness of the representation of lower-frequency processes.**

Section 6.1 will be extended to include the analysis of the RMS values. The title of this subsection will be changed to "**Time series comparison**" and the section will start with a new sentence:

**In order to assess how good North SEAL represents sea level variability on different time scales in comparison with TG, correlations of the time series as well as their RMS differences are analysed.**

In addition, a new paragraph will be added at the end of section 6.1:

**In order to evaluate and quantify the consistency in sea level variability, next to the correlations, RMS differences between altimetry and TG time series are computed and analysed. This is done for all three altimetry dataset for monthly as well as for annual time series, both reduced by potential offsets and trends. In addition, a monthly de-seasoned time series is included in the investigation. Table 3 shows the results from these comparisons. The median RMS are all between three and eight centimeters. North SEAL shows values of 4.9 cm on a monthly scale (without annual signal) and 2.2 cm on an annual scale. With this, it clearly outperforms the two other products on monthly scale. However, for the representation of lower-frequency processes, especially inter-annual variability, the CMEMS product slightly outperforms the other datasets.**

| | North SEAL | CMEMS | SL_cci |
|---|---|---|---|
| RMS of monthly time series | **5.07** | 6.24 | 7.51 |
| RMS of monthly de-seasoned time series | **4.86** | 5.77 | 7.26 |
| RMS of annual time series | 2.20 | **1.94** | 2.31 |

Moreover, the following paragraph will be included in the conclusion:

*__Monthly de-seasoned time series result in a median RMS of the altimetry-TG difference time series of 4.9 cm, while the median RMS in the case of annual mean values is 2.2 cm. These values clearly outperforms other existing altimetry products on a monthly scale, but North SEAL can still be improved for better representing inter-annual variability.__*

Section 6.2: the authors mention that due to the short time periods, the trend uncertainties are of such a magnitude that the trend differences between altimetry and TG, and presumably between different altimetry products as well, are statistically insignificant. Does this mean no robust conclusions can be drawn about whether NorthSEAL outperforms the other altimetry datasets in terms of trends? Since the other comparisons focus on monthly mean timeseries and the seasonal cycle, should NorthSEAL also be preferred over existing altimetry datasets by users interested in interannual sealevel variability and trends? It may help to extend the analysis by looking at both the correlation and the RMSE of monthly and annual mean timeseries.

In fact, the uncertainties related with the trend estimation are very high, not only for the altimetry-based data but also for the TG data. This is mainly due to the relatively short time series and the fact that we take autocorrelation of the input data into account, which increases the uncertainties but make them also more realistic. Thus, based on these data, no final conclusion can be drawn. All three altimetry products provides trend estimates that do not significantly differ from the TG estimates. When comparing the estimated trend uncertainties (which all has been computed with the same metrics), we see similar median values (see Figure below) with slightly better values for North SEAL (1.2 mm/year instead of 1.3 mm/year for SLCCI and CMEMS). This is an indication that North SEAL outperforms the other datasets, but also this is probably not significant.
Thus, you are right: based on the current data, no robust conclusion can be drawn about which of the three altimetry solutions performs best in terms of trend estimation.

In order to evaluate the performance of the datasets in view of inter-annual sea level variability, we will include the analysis of RMSE of time series with different temporal scales in the manuscript (see comment above).

[Figure]

Another thing worth considering is to extend Table 3, which nicely summarizes the comparison of trends, by adding a summary of the key statistics of the comparison of the other metrics as well.

Following your suggestion, we will add the statistics for the validation of the annual amplitudes in the table (now Table 4). In addition, a new table showing the RMS of time series comparison will be added to the manuscript (new. Table 3; also give above).

*The newly derived RMS values will also be discussed in Section 6.2:*

*NorthSEAL shows the lowest absolute mean deviations of the annual cycle of 1.3 cm (CMEMS: 1.8 cm; SL_cci: 1.5 cm)* ***as well as the lowest RMS difference of 1.6 cm, compared to 2.3 cm and 1.9 cm (see Tab. 3).***

Abstract & Conclusions

L12 & 404-405: "It will enable further investigations of ocean processes, sea level projections" & "Moreover, NorthSEAL can also contribute to improve sea level projections" I would suggest some additional explanation on how a regional observational sea-level dataset for the North Sea can directly contribute to improved sea-level projections. Projections do not rely on altimetry observations per se. I can imagine improved process understanding that could result from this dataset could eventually help to improve & interpret projections, or that it may help to better estimate the time of emergence of projected sea-level change above observed variability.

*Some more details on how NorthSEAL contribute to improve sea level projections will be included here. Especially, the indirect influence via the improved process understanding and the model improvements. We will reformulate the complete paragraph on possible applications of North SEAL (see also next comment).*

L405-408: "and it provides an observational basis for planning coastal protection measures. Examples include national and international dike building projects, or an improved understanding whether projects like the Northern European Enclosure Dam NEED (Groeskamp and Kjellsson, 2020) can be a solution for mitigating sea-level related climate change impacts" an improved observational dataset can help to monitor and detect early warning signals to inform adaptation planning, but I wonder what application it finds in studying the feasibility of an idea such as NEED. I would encourage the authors to focus on the direct uses of their dataset in the conclusions instead, including those they already mentioned, such as improved process understanding, model validation and monitoring.

*We agree that NorthSEAL will not be able to provide direct recommendation for NEED. Especially, since the area is too small to include enough information on both sides of the dams. This was only meant as an example for a protection project. With your comment in mind, we will reformulate the complete paragraph on possible applications of NorthSEAL (see also your last comment) to:*

*This enables basin-wide studies of physical processes driving sea level variability, such as the impact of atmospheric wind and pressure forcing**, like it has already done based on the Baltic SEAL dataset, which Passaro et al. (2021) used to study the connection between sea level anomalies in different areas of the Baltic Sea and the North-Atlantic Oscillation. North SEAL will enable similar studies than that of Dangendorf et al. (2014a) based on a consistent dataset for offshore and coastal areas and at a higher spatial scale. Moreover, an improved determination of sea level signatures of coastal currents can be expected. Something that was, until now, only possible using along-track data (e.g. Passaro et al. (2015), Birol and Delebecque (2014)). In addition, the new dataset can help to validate and improve ocean or climate models. Through both, the improved process understanding as well as the improved modelling, NorthSEAL can also be beneficial for sea level projections. It provides an observational basis to better estimate the time of emergence of projected sea level change above observed variability, and for the planning of adaptation and coastal protection measures.***

Other comments

Figures: it is better to use sequential colormaps for non-diverging colorbar ranges. I find this study very helpful for figures: https://doi.org/10.1038/s41467-020-19160-7

Thank you for this comment and the reference. Based on your suggestion we will change the colorbar of most of our figures. For the sequential non-diverging maps we decided to use **YlGnBu**, for diverging one **RdBu**. All used colormaps are color-blind safe.

Title: capitalize 'new'?

Will be changed

L3: 'sea level mean annual amplitudes' could be rephrased to 'amplitudes of the mean annual sea-level cycle'?

Will be changed

L5: 'innovative methods' if possible to specify these methods in a few words it may be worth doing so

We will change that sentence to:

*... and gridded with an innovative least-squares procedure including an advanced outlier detection to a 6-8 km wide triangular mesh.*

L60: 'most probably' can the authors confirm this with the authors of cited paper?

We checked with the authors who confirmed our assumption. The sentence will be changed to

*The altimetry data used was extracted from the global DUACS DT2014 dataset (Pujol et al., 2016) provided by AVISO (personal communication D. LeBars).*

Section 2.1: although the overview is nice, it may be worth checking if all the information in this section is relevant (for example, does the reader need to know the average North Sea temperature?)

We agree that not all given information is relevant for the paper. The information on the temperature will be removed.

Section 2 & 3: can the authors please add how they account for the inverse barometer effect (is it included in the altimetry dataset?)?

The IB effect is corrected by the Dynamic Atmospheric Correction (DAC), which is a combination of IB and high frequent wind effects. We will add some detailed information on this correction in Section 3.1.

*For correcting atmospheric loading and wind effects, the Dynamic Atmospheric Correction (DAC) based on operational atmospheric products is used. Even if corrections based on reanalysis data (i.e. DAC-ERA, Carrere et al. (2016)) may improve the results for the early years, DAC is the only product currently available for the full period under investigation.*

L141: 'refer to' replace by 'see'

will be done

Figure 2: in addition to using a different colorbar range for subplot d), perhaps a different colormap could be used as well to distinguish from the other subplots straight away instead of through the note in the caption.

Will be corrected

Figure 2: in subplot a), a vertical 0 line appears at 0 degrees – might be due to longitude wrapping and the authors could try to fix that; in subplot d) there is a similar vertical line at the left-hand side of the plot.

Thanks for noting this. Will be fixed.

L185: for a detailed description of the methods in this manuscript the reader is referred to a study currently under review, is that problematic?

Meanwhile the paper is published. The reference will be updated.

L208: 'annual amplitudes' may not be readily understood as 'amplitudes of the mean annual sea-level cycle' which I think the authors mean

Will be changed

L224: you compute the amplitude of the mean annual sea-level cycle using the maximum and minimum values in each year. How do you avoid the possibility that the maximum of year 1 is found in December of year 1, and of year 2 in January of year 2, and so effectively the maxima of two years are both found in the same Winter?

The maximum/minimum annual values are actually not obtained individually for each year. Here, we first fit multi-year monthly means, which gives an estimate of the mean annual cycle in form of 12 values (for 12 months). Based on this annual cycle average we select the maximum/minimum monthly value. We agree, if selected separately for each month, we could face such issues.

L227: could the authors please add a few words on why they prefer this approach?

We agree that this can be helpful. We will add:

**It should be highlighted that taking the closest altimetry point to a TG is not always necessarily the highest correlated or most representative observation. Thus,** to match the altimetry sea level data with the TG measurements, we follow the approach of Oelsmann et al. (2021), which only uses the most highly correlated data in the comparison instead of taking the altimetry observation closest to the TG. **Oelsmann et al. (2021) showed that, this approach ensures an increased consistency of along track altimetry and TG observations and enhances the agreement of trends.**

L245: is there a recommended way of interpolating the data on your unstructured grid to a regular grid that end users may prefer to work with?

There are several ways of interpolating the unstructured grids to a regular grid. One way could be the application of the Generic Mapping Tools (GMT) containing the function "surface". It just needs as an input the coordinates and values of the unstructured grid and a definition of the output grid. Please find more information regarding GMT here: https://www.generic-mapping-tools.org/

Another way would be the usage of a python code, which was provided within the frame of the project Baltic SEAL. Baltic SEAL is based on the same unstructured grid used in the current work in the North Sea. Please find the program, related descriptions and information here, http://balticseal.eu/wp-content/uploads/2021/02/BALTIC_SEAL_codes4Novices.zip Please note, the boundaries of the investigation area must be adapted to the North Sea region.

We will add a paragraph on code availability to the manuscript saying:

**A set of Phython codes for novice coders, which has been developed in the frame of the ESA Balic SEAL project, can also be used for NorthSEAL. It provides tools to visualise the data and to convert it to other formats. It is available as a zipped file and can be downloaded from http://balticseal.eu/outputs/.**

Table 2: it may help to add a column with the dimensions of each variable (for example, if the grid is unstructured the longitude and latitude are presumably 2D?)

As we are working with an unstructured grid, the NetCDF files contain only vector data (1D). The dimensions of the unstructured grid are saved in the attribute "nodes". Each file and related variables

contain the same number of nodes/values (12844). This can be easily displayed by calling the UNIX function "ncdump –h *file_name*".

L263-264: not entirely clear what this sentence means, please rephrase

The sentence will be removed from the text.

L282: flagged as what?

Within the gridding procedure all grid nodes at which the SLA data do not fulfill very strict quality rules will be marked by a flag (==1 instead of ==0), which means that the information for this specific grid point should be handled with care. The sentence will be change to

*flagged* **as decreased quality**

L326: does the '23.1%' refer to the percentage of tide gauges at which correlations are lower for NorthSEAL than for SL_cci? How does this align with 'only for a few TG'?

Out of 52 stations, 6 stations perform worse for NorthSEAL than for CMEMS (11.5%). For SL_cci this holds for 12 stations (23.1%). We agree that "few" might be misleading, at least for the comparison to SL_cci. We will change the sentence to

**Only for a minority of TG**, ...

L397-399: the amplitude of the annual cycle is not mentioned here

Thanks for noting this. We will add:

**The mean deviations of the annual amplitude with respect to TG observations is 1.3 cm.**

L403-404: "*This enables basin-wide studies of physical processes driving sea level variability, such as the impact of atmospheric wind and pressure forcing.*" – such studies already exist, so it may be helpful to specify which processes in the North Sea future studies can now investigate using NorthSEAL that previous studies could not?

In fact, probably we will not be able to study additional/new processes. The main point here is, that we do now have a better spatial resolution and a consistent dataset for the entire area, including open ocean and coastal areas. To emphasis this, the manuscript will be changed to:

**NorthSEAL will enable to study those processes based on a consistent dataset for offshore and coastal areas and at a higher spatial scale as previous studies (e.g. Dangendorf et al., (2014a)).**

---

## Author Comment (AC2)

We thank the reviewer for the valuable feedback. Their comments and suggestions helped to improve the manuscript. Below, we provide an answer to each point and explain the corresponding changes adopted in the manuscript. The original review comments are copied from the report (in black) and our answers are in blue. Manuscript changes are given in *italics.*

Anonymous Referee #2

- The influence of wind on sea level in the North Sea is large because it is shallow (Dangendorf et al. 2014). Most of the monthly time scale sea level variability is wind driven. Therefore I am surprised that there is no more discussion of the Dynamic Atmospheric Correction. Is the DAC-correction here the same as for the other two altimetry products? Why use a DAC based on atmospheric analysis? Using a DAC based on the ERA-interim reanalysis showed an improvement (Carrère et al. 2016) and ERA5 would improve further.

DAC is the only atmospheric loading correction that is freely available for the full period under investigation. We agree that DAC-ERA would be a better choice and would help to improve the results, especially for the early years. Unfortunately, DAC-ERA is only available until end of 2015. So we would miss more than three years. For that reason, we decide for the ECMWF driven DAC product instead of mixing different corrections.
*We will include some more information on that in Section 3.2 (along-track data preprocessing). In addition, Dangendorf et al. (2014) will be cited in Section 2.1.*

- I am curious if there are plans to keep this dataset up to date in the future. That could be mentioned somewhere.

We don't plan to extend the dataset as an operational service. However, we will update the dataset in irregular intervals to keep track of the long-term evolution in this area. This might also include product updates due to improved altimeter reprocessing (e.g. TOPEX) or improved correction models (e.g. DAC-ERA5).
*A sentence on these plans will be put in the outlook.*

- I strongly advise the authors to also share the code used to make the analysis in this manuscript. Especially since NorthSeal is not on a standard rectilinear grid the use of the data by other people would be greatly simplified with an example.

Thank you for this suggestion. In fact, we already published some useful software tools in the frame of the Baltic SEAL project. This includes a python code to interpolate the unstructured data to a regular grid. Baltic SEAL. Baltic SEAL is based on the same unstructured grid used in the current work in the North Sea. The software can easily be transferred to a different regions when the boundaries of the investigation area are adapted to the new region.
Please find the program, related descriptions and information as well as some examples here: http://balticseal.eu/wp-content/uploads/2021/02/BALTIC_SEAL_codes4Novices.zip.

*We will add a paragraph on code availability to the manuscript saying "A set of Phython codes for novice coders, which has been developed in the frame of the ESA Balic SEAL project, can also be used for North SEAL. It provides tools to visualise the data and to convert it to other formats. It is available as a zipped file and can be downloaded from http://balticseal.eu/outputs/."*

- l.292: It is interesting to compare with Wahl et al. 2013. As additional potential source of discrepancy you could also mention that their region was different, it extended further into the Channel and they found much smaller trend in the Channel (1.32 +-1.11) than in the inner North Sea (4.59 +-1.82). And the GIA uncertainty is also large.

Thank you very much for this information. *We will include it in the manuscript.*